# Comparison of enteric methane yield and diversity of ruminal methanogens in cattle and buffaloes fed on the same diet

P. K. Malik[1]*, S. Trivedi[1], A. Mohapatra[1], A. P. Kolte[2], V. Sejian[3], R. Bhatta[1], H. Rahman[4]

1 Bioenergetics and Environmental Science Division, ICAR-National Institute of Animal Nutrition and Physiology, Bangalore, Karnataka, India, 2 Animal Nutrition Division, ICAR-National Institute of Animal Nutrition and Physiology, Bangalore, Karnataka, India, 3 Animal Physiology Division, ICAR-National Institute of Animal Nutrition and Physiology, Bangalore, Karnataka, India, 4 International Livestock Research Institute, South Asia Regional Office, New Delhi, India

☯ These authors contributed equally to this work.
* malikndri@gmail.com

**Data Availability Statement:** All relevant data are within the manuscript and its S1 Fig and S1 File and S1 Table.

## Abstract

An *in vivo* study was conducted to compare the enteric methane emissions and diversity of ruminal methanogens in cattle and buffaloes kept in the same environment and fed on the same diet. Six cattle and six buffaloes were fed on a similar diet comprising Napier (*Pennisetum purpureum*) green grass and concentrate in 70:30. After 90 days of feeding, the daily enteric methane emissions were quantified by using the $SF_6$ technique and ruminal fluid samples from animals were collected for the diversity analysis. The daily enteric methane emissions were significantly greater in cattle as compared to buffaloes; however, methane yields were not different between the two species. Methanogens were ranked at different taxonomic levels against the Rumen and Intestinal Methanogen-Database. The archaeal communities in both host species were dominated by the phylum *Euryarchaeota*; however, *Crenarchaeota* represented <1% of the total archaea. Methanogens affiliated with *Methanobacteriales* were most prominent and their proportion did not differ between the two hosts. *Methanomicrobiales* and *Methanomassillicoccales* constituted the second largest group of methanogens in cattle and buffaloes, respectively. *Methanocellales* (*Methanocella arvoryza)* were exclusively detected in the buffaloes. At the species level, *Methanobrevibacter gottschalkii* had the highest abundance (55–57%) in both the host species. The relative abundance of *Methanobrevibacter wolinii* between the two hosts differed significantly. *Methanosarcinales*, the acetoclastic methanogens were significantly greater in cattle than the buffaloes. It is concluded that the ruminal methane yield in cattle and buffaloes fed on the same diet did not differ. With the diet used in this study, there was a limited influence (<3.5%) of the host on the structure of the ruminal archaea community at the species level. Therefore, the methane mitigation strategies developed in either of the hosts should be effective in the other. Further studies are warranted to reveal the conjunctive effect of diet and geographical locations with the host on ruminal archaea community composition.

**Funding:** This work was supported by the International Livestock Research Institute (ILRI), Nairobi, Kenya through window-III program of Department of Agricultural Research and Education (DARE), New Delhi, India.

**Competing interests:** No authors have competing interests.

## Introduction

Methane is the second most significant greenhouse gas in the atmosphere [1]. The present atmospheric concentration of methane is about 1889 ppb which is increasing at an average rate of 10 ppb per year [2]. Enteric fermentation with an annual emission of 87–97 Tg, remains one of the largest sources of methane in the agriculture sector [3]. The contribution of global cattle and buffaloes to the annual enteric methane emission is 77 and 13%, respectively [4]. India has about 13% of the global population of cattle and 53% of the global population of buffaloes [5] and these account for 4.92 and 2.91 Tg of annual global enteric methane emission from the respective species [6]. These two major bovine species are aggregately responsible for over85% of total enteric methane emission in India. The cattle and buffaloes contribute 48 and 49% to the total annual milk production (198 million metric tonnes) in India [7]. Methanogenesis is a major sink for $H_2$ in the rumen. However, methane has an embodied energy of 39.5 kJ/l [8], resulting in the methane emitted by cattle accounting for 2–12% of gross energy intake [9].

Though the ruminal methanogens are not as diverse as bacteria; nevertheless, the contrast substrate requirement for different categories makes this community complex. Hence, understanding the ruminal archaea community is crucial for devising effective methane mitigation strategies. Host impact on the ruminal microbiota establishment has been reported previously [10–14]. Environmental conditions have also been reported to influence the ruminal archaea community [15]. *Methanobrevibacter gottschalkii*, *Methanobrevibacter ruminantium* clades, *Methanosphaera* sp. and two *Methanomassiliicoccaceae*-affiliated groups with variable abundance, dominate the archaeal community in all animal species globally [16]. Due to the variable physical and chemical characteristics of feed, diet remains a major determinant that shapes the microbial community in the host animal [16]. In contrast, some remarkable differences were reported in the dominance of ruminal archaea in host animals [17–20].

From these studies, it remains uncertain whether host, diets, or geographical locations lead to a difference in the archaeal community. Therefore, we hypothesized that there would be no difference in the methane yield or methanogens demographics between cattle and buffaloes when fed on the same diet. To disallow any difference in the archaeal community due to the diet and environmental conditions, a study was designed to compare the methane yield and archaeal community composition between cattle and buffaloes fed on the same diet composed of Napier grass and concentrate.

## Materials and methods

### Animals and feeding

The experiment was conducted at the Livestock Experimental Station of the ICAR-National Institute of Animal Nutrition and Physiology, Bangalore situated in the Indian Deccan Plateau at an average elevation of 900 m at 12.97˚N and 77.56˚E. This study was carried out in strict accordance with the Protocols for the Animal Experiments of ICAR-National Institute of Animal Nutrition and Physiology, Bangalore, India. The study was approved by the Institutional Animal Ethics Committee (approval no. NIANP/IAEC/1/2020/5).

An *in vivo* study to compare the enteric methane emission and the composition of ruminal methanogens community was conducted in six adult male cattle (BW 538±23.3 kg) and six adult male buffaloes (BW 284±14.5 kg). To disallow any difference in the diversity due to geographical region, both the cattle and buffaloes were kept in the same environmental conditions during the entire experimental period. The average minimum and maximum temperatures during the experimental period were 21.8 and 35.8˚C; while the average humidity was 57%.

The animals were housed group-wise in the tail to tail orientation in a half-open shed (cemented wall up to 1.8 m and then iron wire mesh 1.75 m) having a provision for the individual housing and feeding. The animals were dewormed orally with fenbendazole (5mg/kg BW) before the beginning of the experiment. Another important factor that affects the composition of the methanogen community in the rumen is diet variation; hence, in the present study, animals of both species were fed on a same diet comprising Napier (*Pennisetum purpureum*) green grass and concentrate in the ratio of 70:30. The concentrate mixture was prepared using maize grain (320 g/kg), Soybean meal (130 g/kg), groundnut cake (120 g/kg), wheat bran (400 g/kg), mineral mixture (20 g/kg), and salt (10 g/kg). Experimental animals were offered the feed daily at 09.00h; while clean drinking water was accessible to the animals throughout the day. The dried and ground feed and concentrate samples were analyzed for crude protein (Nx6.25) and ash content as per AOAC [21]. However, the fibre constituents such as neutral detergent fibre (NDF) and acid detergent fibre (ADF) were determined according to Van Soest et al. [22]. The chemical composition (% dry matter basis) of the green fodder and concentrate is given in the S1 Table.

## Methane emission

After 90 days of feeding, the daily enteric methane emission in cattle and buffaloes was quantified using the sulfur hexafluoride ($SF_6$) tracer technique [23] for the consecutive 7 days. During the methane measurement study, the dry matter intake for the individual animal was recorded by considering the amount of feed offered and refusals. The permeation tube made from 8.5 mm diameter brass rod, was 34 mm long, fitted with a Swagelok nut, and bored with a 4.8 mm blind hole about 30 mm deep. A Teflon septum (0.24 mm PTFE) acts as a permeable membrane and supported against internal pressure by a stainless steel frit (3/8" OD, 2 μm pore size) and held in place by the nut, whose 7.0 mm diameter hole provides the permeation window. A nylon washer has been included between the Teflon and polished brass face. Tubes were charged at liquid nitrogen temperature at which 747±55.48 mg SF6 (99.9% pure) was frozen into the tube from direct syringe injection. Charged tubes were retained in an incubator at 39°C and monitored through daily weighing. Tubes were calibrated by weighing (Denver TP214, Germany) for 60 days. The release rate was calculated by linear regression of the tube weights obtained during the calibration period. On completion of the calibration, tubes were inserted into the cattle and buffaloes rumen seven days the commencement of methane measurement. The $SF_6$ release rates from the tubes were 3.39±0.56 mg/d (mean±SD). Keeping the importance of background sample in view, the PVC canister was connected to a nylon tube, capillary tube (Supelco, 56712-U, ID 1/16) and Quick connectors (Swagelok, B-QC4-D-200) and assembled as per Williams et al. [24]. The canister for background sample was hung on the ventilated iron wire mesh fixed in the cemented wall above a height of 1.8 m from the ground. Before analysis, both breath and background samples in canister were diluted (2.47–3.51 folds) with high purity $N_2$ gas and successive sub-samples were collected in a *Hamilton* syringe (gastight 1001, 1 ml). The diluted gas samples were injected into the gas chromatograph (GC 2010 plus, Shimadzu, Japan) for the estimation of methane and sulfur hexafluoride gas concentrations by flame ionization detector (FID) and electron capture detector (ECD), respectively. Following chromatographic conditions were maintained for SF6 analysis: inlet temperature 100°C, column temperature 40°C, detector temperature 250°C, airflow rate 400 ml/min, hydrogen flow rate 40 ml/ min and nitrogen flow rate 30 ml/min; while for $CH_4$ analysis the followings conditions were held: inlet temperature 100°C, column temperature 60°C, detector temperature 150°C, airflow rate 400 ml/min, hydrogen flow rate 40 ml/min and nitrogen flow rate 30 ml/min. The physical dilution with N2 was mathematically accounted for

using the equation of Lassey et al. [25] with adjustment for local elevation and atmospheric pressure.

$$[G_S] = \frac{90 - \tau_f}{\tau_e - \tau_s} x [G_A]$$

Where, $G_S$ is the calculated concentration of methane (ppm) or SF6 (ppt) at average atmospheric pressure of 90 kPa at an elevation of 920 m. $\tau_f$ (kPa) was the final vacuum in canister after the addition of N2, $\tau_s$ (kPa) was the vacuum in the canister after the sample collection, $\tau_e$ was the vacuum in the evacuated canister before use, $G_A$ was the concentration of methane (ppm) or SF6 (ppt) in the sample presented to the GC.

The daily enteric methane emission was calculated using the formula of Moate et al. [26].

$$R_{CH4} = R_{SF6} \frac{[CH_4]_M - [CH_4]_{BG}}{[SF_6]_M - [SF_6]_{BG}} \times \frac{MW_{CH_4}}{MW_{SF_6}} \times 1000$$

$R_{CH4}$ is daily $CH_4$ output (g/d); $R_{SF6}$ is $SF_6$ release rate from the permeation tube; $MW_{CH4}$ is the molecular mass of $CH_4$; $M_{SF6}$ is the molecular mass of $SF_6$.

To compare the emission between cattle and buffaloes on a uniform dry matter intake basis, the methane yield (MY, g/kg DMI) was calculated using mean methane emission divided by the mean dry matter intake (DMI) over the measurement period.

The data generated were analyzed using IBM SPSS Statistics for Windows, (Version 21.0. Armonk, NY: IBM Corp.) and following the analysis of variance (ANOVA). The difference between the means was compared by Tukey's method and considered significant at P≤0.05.

## Ruminal fluid collection

At the end of methane measurement trial, the ruminal fluid samples (~45 ml) were collected from the individual animal using a nylon stomach tube (length 2 m) connected to a vacuum pump (Mityvac 8000) having sample collection vessel. The collected ruminal fluid samples were filtered through double layer of muslin cloth and placed on the ice before transporting to the laboratory. Each sample was divided into three equal subsets. The first subset of ruminal fluid was processed for the ammonia-N and volatile fatty acid (VFA) estimation. For the estimation of ammonia-N, about 2–3 drops of saturated $HgCl_2$ was added to the supernatant obtained after centrifugation, whereas metaphosphoric acid (25%) in 1:4 (v/v) was added to the supernatant of ruminal fluid for VFA estimation The processed samples were stored at -20˚C until further processing. For protozoal enumeration, an equal volume of formaldehyde (37%) was added to the second subset and processed for the counting. The third subset of ruminal fluid was preserved at -80˚C till the process for DNA isolation.

## Estimation of VFA and ammonia-N

The VFA concentration in the ruminal fluid samples was determined as per Filipek and Dvorak [27] using a gas chromatograph (Agilent, 7890B) with slight modifications. An FFAP capillary column (CP7485, 25 m × 0.32 mm × 0.30 μm, Agilent Technologies) was used. The nitrogen was used as carrier gas–flow rate 2 ml/min. The FID detector with the following conditions was used for VFA estimation: temperature programme 59–250˚C (20˚C/min, 10 min), injector– 230˚C, detector– 280˚C. The injector was equipped with a glass liner containing glass wool to separate dirt particles from the sample. The samples were dosed by a G4513A automatic liquid sampler at an injection size of 1 μl using the split method with a 20:1 splitting ratio. The analysis time was approximately 16.7 min.

The concentration of ammonia-N in the samples was determined following the procedure of Conway [28]. Briefly, 1 ml of mixed boric acid indicator was pipetted in the inner chamber of the disc; while an equal volume of saturated sodium carbonate was placed in the outer chamber. About 1 ml of strained ruminal fluid was pipetted just opposite the sodium carbonate in the outer chamber. The disc was covered with the lid and gently rotated before incubation for two hours at room temperature. After completion of incubation, contents of inner chamber were titrated against 0.01N sulfuric acid till colour turned to pink. Ammonia-N was determined using following formula.

$$Ammonia - N \ (mg/dl) = ml \ of \ 0.001N \ H_2SO_4 \ x \ 14$$

## Protozoal enumeration

The enumeration of protozoa in ruminal fluid was performed using a Neubauer counting chamber according to the method described previously [29]. The ruminal protozoa based on the gross morphology and distribution of cilia over the body surface were categorized into *Entodiniomorphs* or *Holotrichs* [30].

## DNA isolation

The frozen ruminal fluid samples were thawed at room temperature, centrifuged at 1000g for 5 min to allow the sedimentation of dissolved micro feed particles, and supernatants were collected. About 2 ml supernatant was taken into an *Eppendorf* tube and centrifuged at 4˚C, 13400x g for 10 minutes. A thick pellet obtained by the centrifugation was retained while removing the supernatant carefully. Subsequently, 1 ml lysis buffer was added to it and dissolved pellet through gentle pipetting. The mixed content was transferred to a 2 ml sterile screw cap tube contained 0.5 g (0.1 mm) pre-sterilized zirconia beads (BioSpec, USA). The repeat bead beating plus column method [31] was used for the genomic DNA isolation in the present study. The QIAamp DNA mini kit (Qiagen, Germany) was used as per the manufacturer's instructions. The quality of the genomic DNA was checked using 0.8% agarose gel electrophoresis; while the DNA concentration was confirmed with Qubit 4.0 (Invitrogen).

## Library construction and sequencing

Genomic DNA samples were processed for the preparation of amplicon libraries and sequencing. Amplicon libraries were prepared using the Nextra XT kit (Illumina Inc.). Archaea specific primers *Arch-344F* 5' ACGGGGYGCAGCAGGCGCGA 3' [32] and *Arch-806R* 5' GGAC TACVSGGGTATCTAAT 3' [33] were used for the amplification. Illumina adapters i5 and i7 were added to the primers for generating the amplicons. Amplicon libraries were purified by AMPureXP beads (Beckman Coulter Life Sciences, USA) and analyzed on 4200 Tape Station system (Agilent Technologies, USA) using D1000 Screen Tape station. About 10–20 pM of each library was loaded onto the MiSeq platform for cluster generation and sequencing.

## Bioinformatics analysis

Raw amplicon sequence reads generated from Miseq were processed using DADA2 V1.16 [34] in R V4.0.2. Raw reads quality was assessed using function plot Quality Profile followed by dereplication, denoising, and merging of the paired reads. The Truncation parameter was set to 295 and 260 nucleotides for the forward and reverse reads, respectively. Chimeras were removed from the filtered reads using function removeBimeraDenovo by implementing the consensus method with minFoldParentOverAbundance. DADA2 compatible reference fasta file for taxonomy assignment was generated using Rumen and Intestinal Methanogen-DB

(RIM-DB) [35] by an in-house python script. Further, taxonomy classification was performed on the reads using assign taxonomy function against the RIM-DB. An annotated taxonomy table from DADA2 was imported into phyloseq package V1.26.1 [36] in R for further downstream analysis. Low abundance OTUs (operational taxonomic unit) were pruned and samples were rarefied to the lowest read numbers to examine the archaeal diversity measures. The rarefaction curve was plotted using the rarefy function from vegan package V2.0–7 [37]. Alpha diversity measure was estimated by the Shannon index and *post hoc* comparison was performed using pairwise Wilcoxon ranksum test. To test the multivariate homogeneity of group dispersion, betadisper function from vegan package was implemented. Further, principal coordinate analysis was performed based on the Bray-Curtis dissimilarity matrix and *post hoc* comparison was done using Permutational Multivariate Analysis of Variance (PERMANOVA) by adonis function in Vegan V2.0–7. The OTUs abundance at different taxonomic ranks was studied between two host species cattle and buffaloes and the relative abundance plots were generated using ggplot2 [38]. The OTU count data was normalized and differential abundance significance was tested at different taxonomic ranks using Wald parametric test with Benjamin-Hochberg correction from DESeq2 [39]. Core microbiome analysis was also performed using package microbiome V1.4.1 [40] in R with a minimum prevalence and detection threshold of 50% and 0.01, respectively.

## Data availability statement

The archaeal metagenome sequencing reads from the experiment are accessible at the NCBI Sequence Read Achieve (SRA; https://www.ncbi.nlm.nih.gov/subs/sra) accession numbers SAMN16378101- SAMN16378111 under BioProject PRJNA667560. The OTUs abundance and taxonomical assignment data are available in S1 File.

## Results

### Dry matter intake and methane emission

*In vivo* study revealed that the enteric methane emission (g/d) was significantly greater ($p < 0.001$) in cattle as compared to the buffaloes. Similarly, a greater ($p < 0.001$) dry matter intake was also recorded in the cattle than the buffaloes (10.5 Vs. 6.86 kg/d). However, the comparison of methane yield (g/kg DMI) calculated using daily methane emission and mean DMI revealed a non-significant difference (P = 0.519) in enteric methane yield (Table 1) between cattle (13.4 g/kg DMI) and buffaloes (13.5 g/kg DMI) fed on the same diet.

### Rumen fermentation and protozoal concentration

There was no difference in mean concentrations of either ammonia-N or total volatile fatty acid (TVFA) in the ruminal fluid of cattle and buffaloes. Similarly, the individual fatty acid concentration except iso-butyrate was also comparable between the two host species (Table 1) fed on a similar diet consisting of Napier grass and concentrate. The numbers ($x10^6$/ml) of total protozoa and *Entodiniomorphs* were lesser ($p < 0.05$) in cattle than the buffaloes. However, the *Holotrichs* numbers ($x10^6$/ml) were comparable between the two host species.

### Effect of the host on methanogens community

A total of 2,628,682 archaeal raw reads with an average of 238,971 reads per sample were generated from the study. After quality filtering and chimera removal, a total of 827,901 reads were retained for further analysis (S1 File). Taxonomy classification of reads at $\geq$97% similarity against the RIM-DB clustered produced a total of 3,924 archaeal OTUs. The rarefaction

**Table 1. Comparison of ruminal methanogenesis and mean concentrations of ammonia and total volatile fatty acids as well as molar proportions of principal volatile fatty acids in ruminal fluid of cattle and buffaloes.**

| Attributes | Cattle | Buffaloes | SEM | P value |
|---|---|---|---|---|
| Ammonia N (mg/100 ml) | 8.63 | 9.10 | 0.652 | 0.739 |
| TVFA (mM/l) | 93.6 | 57.3 | 12.45 | 0.152 |
| **Individual VFA (molar proportion)** | | | | |
| Acetate | 60.7 | 61.4 | 0.730 | 0.639 |
| Propionate | 15.2 | 14.7 | 0.285 | 0.391 |
| Butyrate | 19.2 | 18.9 | 0.566 | 0.826 |
| Iso-butyrate | 0.91[a] | 1.05[b] | 0.033 | 0.025 |
| Valerate | 2.52 | 2.75 | 0.093 | 0.241 |
| Isovalerate | 1.41 | 1.08 | 0.136 | 0.243 |
| **Protozoa** | | | | |
| Total (x10$^6$/ml) | 23.6[a] | 36.1[b] | 2.73 | 0.014 |
| *Entodiniomorphs* (x10$^6$/ml) | 23.4[a] | 35.8[b] | 2.74 | 0.014 |
| *Holotrichs* (x10$^6$/ml) | 0.165 | 0.226 | 0.101 | 0.319 |
| **Methane** | | | | |
| Methane emission (g/d) | 141 | 93.1 | 3.350 | <0.001 |
| Methane Yield (g/kg DMI) | 13.4 | 13.5 | 0.043 | 0.089 |

TVFA- total volatile fatty acid, VFA- volatile fatty acid; SEM- standard error of mean; DMI- dry matter intake

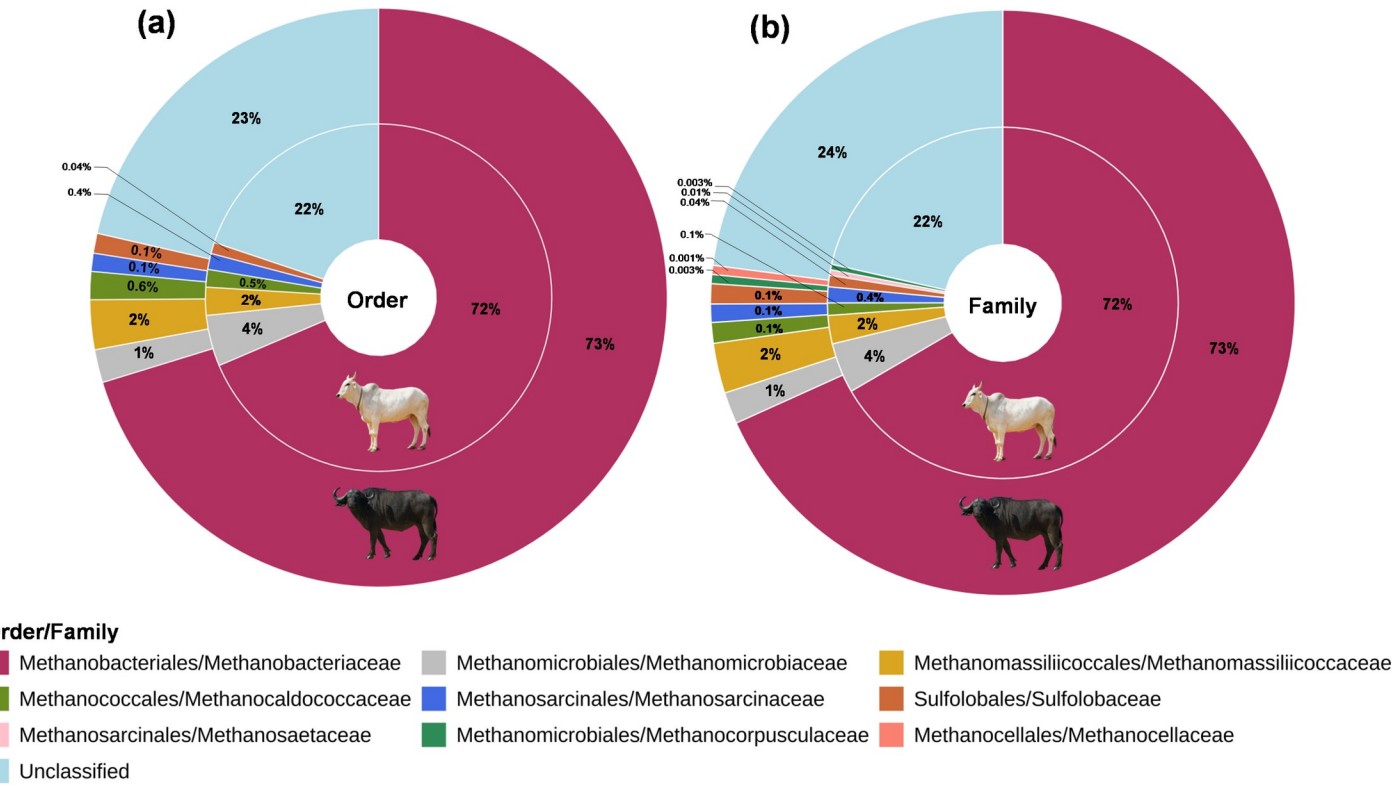

**Order/Family**

- Methanobacteriales/Methanobacteriaceae
- Methanomicrobiales/Methanomicrobiaceae
- Methanomassiliicoccales/Methanomassiliicoccaceae
- Methanococcales/Methanocaldococcaceae
- Methanosarcinales/Methanosarcinaceae
- Sulfolobales/Sulfolobaceae
- Methanosarcinales/Methanosaetaceae
- Methanomicrobiales/Methanocorpusculaceae
- Methanocellales/Methanocellaceae
- Unclassified

**Fig 1.** Ruminal archaea community composition in cattle and buffaloes at (a) the order level and (b) family level. The inner-circle represents the methanogens distribution in cattle; while the outer circle represents the distribution in buffaloes.

curve prepared from the archaeal OTUs confirm the adequate coverage of the diversity of archaea (S1 File and S1 Fig). All filtered reads in the present study were affiliated to the archaea. Taxonomic annotation of OTUs revealed that the ruminal archaea community was dominated by the phylum *Euryarchaeota* in both cattle (98%) and buffaloes (97%). In this study, the phylum *Crenarchaeota* representation was only <1%.

About 22–24% of the reads irrespective of the host at order and class levels remain unclassified. Methanogens belonging to six orders were identified in the cattle; while there were seven orders in the buffaloes. The Methanogens associated with the *Methanocellales* were exclusively identified in the buffaloes. The methanogens affiliated to the *Methanobacteriales* were dominant in both the host and represented 72–73% of the total ruminal archaea (Fig 1A). There was no difference (P = 0.872) in the distribution of *Methanobacteriales* between the cattle and buffaloes. In cattle, *Methanomicrobiales* were the second most abundant methanogens (3.75%); whilst they constituted only 0.85% of the total archaea in buffaloes. In buffaloes, *Methanomassillicoccales* represented the second largest group of methanogens. However, the distribution of either *Methanomicrobiales* (P = 0.245) or *Methanomassillicoccales* (P = 0.872) did not differ between the two host species. Though the *Methanosarcinales* were distributed at a lower frequency in both cattle and buffaloes; nevertheless, their abundance was significantly higher (p<0.001) in the cattle as compared to buffaloes. Morphologically and physiologically distinguished archaea belong to the order *Sulfolobales* (*Crenarchaeota*) were also identified in both host species.

Methanogens belonging to a total of nine families were identified in this study (Fig 1B). All the orders were represented by one family; however, the orders *Methanomicrobiales* and *Methanosarcinales* in both host species were characterized by the two families of each. Most of the orders had only single taxa classified at the family level and thus precisely represented the same relative abundance at both hierarchy levels (order and family). *Methanomicrobiaceae* and *Methanocorpusculaceae* were affiliated to the *Methanomicrobiales*; while *Methanosarcinaceae* and *Methanosaetaceae* were classified within the order *Methanosarcinales*. At the family level, the distribution of *Methanosarcinaceae* methanogens was only significantly higher (p<0.001) in cattle than in buffaloes (Fig 1B and S1 File).

In the present study, 14 genera of the methanogens were identified in cattle; while archaea associated with 12 genera were identified in the buffaloes. The *Methanobrevibacter* irrespective of the host species was most prominent genus and represented 66–68% of the total archaea in the rumen. However, their distribution between cattle and buffaloes was comparable. Methanogens affiliated to *Methanobacterium* and *Methanomicrobium* were the second and third largest genus in cattle and aggregately constituted 7.3% of the archaea. On the other hand, *Methanosphaera* and *Methanobacterium* were the second and third largest genus of methanogens in buffaloes, respectively, and aggregately constituted about 5% of the ruminal archaea. The *Methanosaeta* and *Group 8* methanogens were distributed at a low frequency and exclusively detected in the rumen of cattle. On the other hand, *Methanocella* were detected in the buffaloes but remain undetected in cattle (Fig 2 and S1 File).

In the present study, 25 species of the methanogens were reported in the rumen of cattle; while there were 20 species detected in the buffaloes. Among the species, *Methanobrevibacter gottschalkii* had highest abundance (55–57%) in both the host species. There was no significant (P = 0.869) difference in the distribution of *Methanobrevibacter gottschalkii* between the cattle and buffaloes. The relative abundance of *Methanobrevibacter wolinii* despite the limited representation in the rumen varied significantly (P = 0.049) between the two host species. Their distribution was significantly greater in cattle as compared to the buffaloes. Though the abundance of each of *Methanobacterium formicicum*, *Methanobacterium beijingense*, *Methanobacterium sp*., *Methanobacterium movens*, *Group 8*, *Methanosarcina barkeri* and

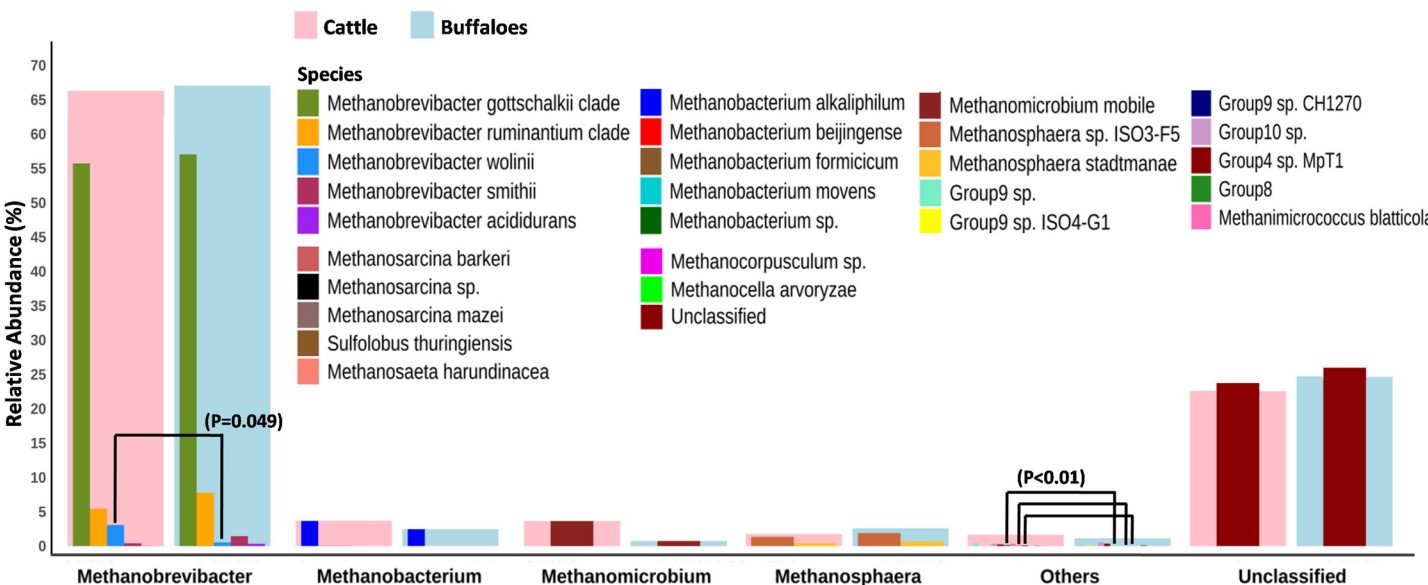

**Fig 2. Ruminal archaea community composition at genus and species levels in cattle and buffaloes.** Each genus is represented by larger bars that are underlaid on the smaller bars representing the abundance of all the species affiliated to the corresponding genus.

*Methanosarcina sp.* was very limited, they were all, nevertheless, exclusively detected in the rumen of cattle. Similarly, *Group 9 sp. CH1270* and *Methanocella arvoryzae* methanogens were exclusively identified in the buffaloes (Fig 2 and S1 File).

## Spatial components of biodiversity in the hosts

The comparative archaeal distribution in cattle and buffaloes studied using Shannon index (alpha diversity) and Bray-Curtis (beta diversity) is presented in Fig 3. There was no significant difference (P = 0.66) in the alpha diversity between host species. However, the overall ruminal methanogens diversity between cattle and buffaloes was significant (P = 0.015). Betadisper test was not significant, indicating a multivariate homogeneity of host dispersion (P = 0.648). Bray-Curtis beta diversity analysis indicated a significant difference in ruminal methanogens community composition between the host species (P = 0.01).

## Core methanogens analysis

The methanogens constituted the core archaeome at genus level were similar between cattle and buffaloes. However, *Methanomicrobium* was exclusively identified as a part of the archaeal core microbiome in cattle (Fig 4A and 4B). At the species level, *Methanobrevibacter gottschalkii clade*, *Methanobrevibacter ruminantium clade*, *Methanobacterium alkaliphilum*, and *Methanosphaera sp. ISO3-F5* composed the core methanogens microbiome in both host species. The methanogens such as *Methanomicrobium mobile* and *Methanobrevibacter wolinii* were exclusively present in the archaeal core microbiome in cattle (Fig 4C). On the other hand, *Methanobrevibacter smithii* exclusively represented the methanogens core microbiome in buffaloes (Fig 4D).

## Discussion

There is a dearth of literature comparing the methane emissions between cattle and buffaloes fed on the same diet and maintained under similar environmental conditions. Our results revealed that the daily methane emissions were significantly greater in cattle than in buffaloes.

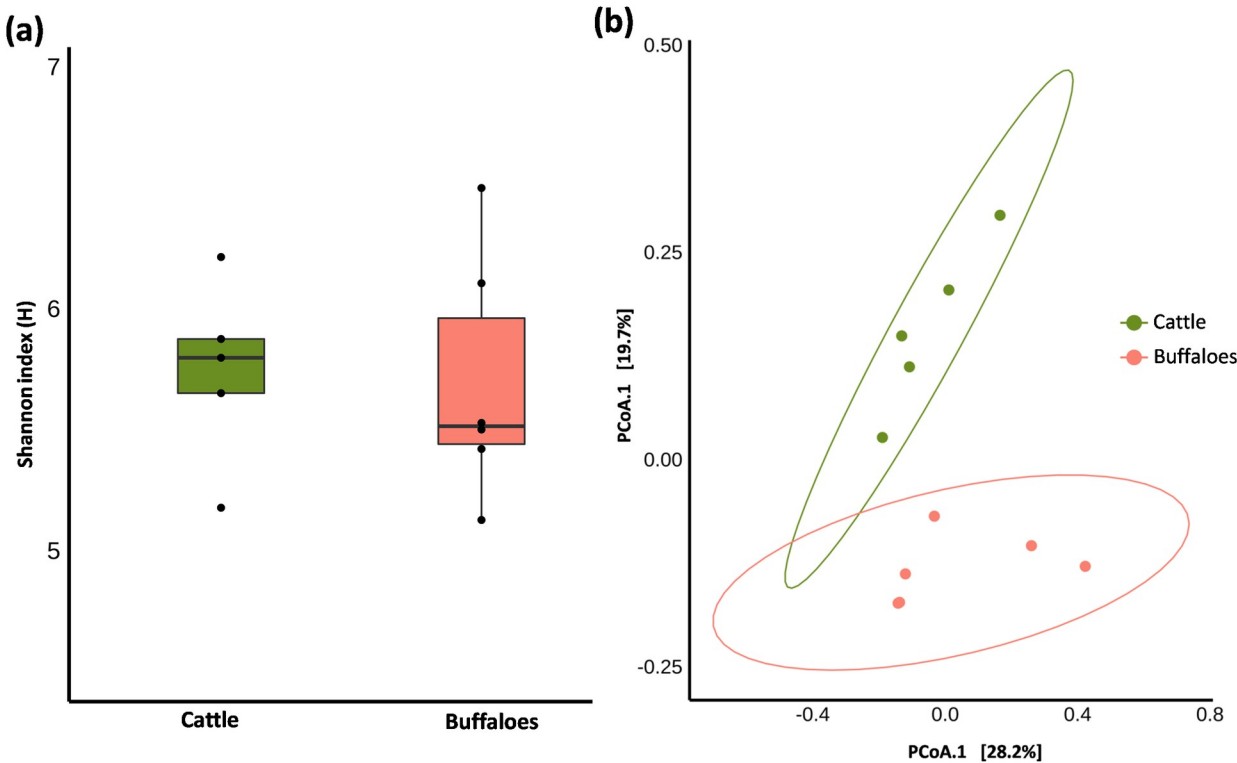

**Fig 3.** (a) Alpha diversity and (b) Beta diversity of the ruminal methanogens in cattle and buffaloes.

However, this difference in daily methane emissions was attributed to the significantly greater dry matter intake and body weight in cattle (BW 538 kg; 10.5 kg DMI) as compared to buffaloes (BW 284 kg; 6.86 kg DMI). These results are in good agreement with a previous study [41], where a significant difference in enteric methane emission due to higher dry matter intake and body weight between cattle and buffaloes was reported. The methane yield (g/kg DMI) between cattle and buffaloes in this study was not affected by the host species on the same diet and it was within the acceptable range of 12–30 g/kg DMI [42]. However, methane yield in both cattle and buffaloes were lower than reported by Charmley et al. [43] for high forage diet (>70%). The reason for the lower methane yield in this study could be attributed to

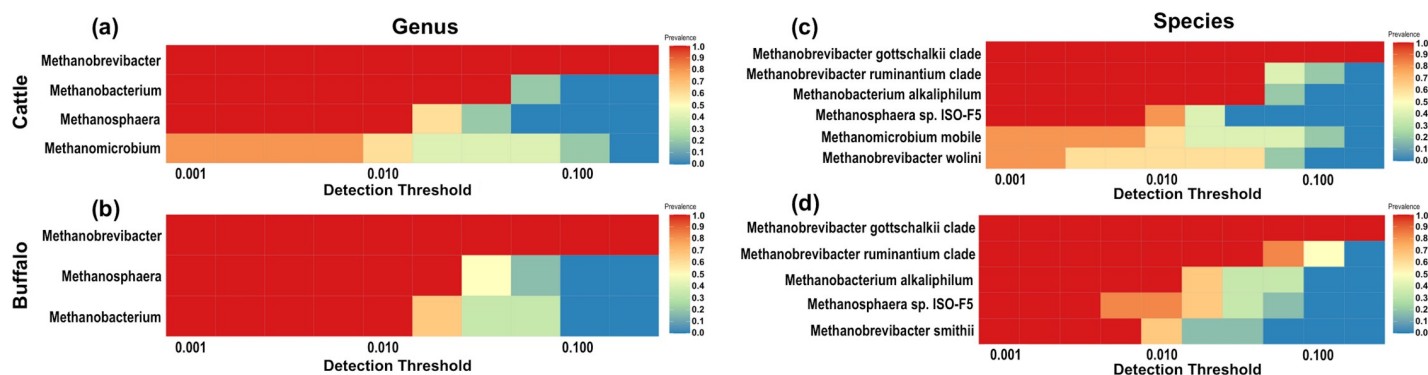

**Fig 4.** Ruminal archaea representing the core microbiome at 50% minimum prevalence in (a) Cattle at genus level (b) buffalo at genus level (c) cattle at species level (d) buffaloes at species level. The colour gradient indicates variability in prevalence.

the presence of tannins and saponins inhibitors in the Napier grass [44, 45], which are well known for lowering methane emission.

This study did not detect any statistical difference in the concentrations of ammonia-N or TVFA in the ruminal fluid of cattle and buffaloes, although there were some substantial numerical differences. Similarly, except for iso-butyrate, there were no differences in the molar proportions of individual volatile fatty acids in the ruminal fluid of cattle and buffaloes. The greater dry matter intake in cattle as compared to that of buffaloes can be attributed to the greater body weight. Both total protozoa and *Entodiniomorphs* numbers were significantly smaller in cattle as compared to buffaloes (Table 1). This is in agreement with the findings of Jabari et al. [46].

Archaea belong to the domain *Euryarchaeota* represents 3–5% of the rumen microbiota [16, 47]; nevertheless, they have a major role in maintaining a low $H_2$ pressure within the desirable limits [48, 49]. It has been reported that the rumen microbiota could be affected by the host [16, 50, 51], diet, and geographical locations [16]. A remarkable difference in the rumen microbiota composition was observed between hosts [52] and breeds fed on a similar diet [53]. In a global study, Henderson et al. [16] demonstrated that diet and host as compared to the geographical locations have a major impact on the rumen microbiome. Their study in cattle from Ontario and Prince Edward islands demonstrated the influence of geographical locations and diet on the diversity of methanogens [16]. Their findings confirmed that the ruminal archaea community are less diversified than the bacteria and a major fraction of the archaea remain invariable across geographical locations [54]. More recently, Trivedi et al. [55] established that the geographical region and host influence the archaeal community composition. The multiple factors (diet, host, geographical locations) were confounded in most of these studies and due to the variability of more than one factor, it is very difficult to clearly separate the impact of host or diet or geographical locations on the ruminal methanogens community structure. To overcome the confounding effect, the present study reports the effect of host species on the ruminal methanogens diversity; while two other important factors diet and geographical locations were kept constant. While the present study was conducted with a single diet, and we speculate results might differ with other diets, our study indicates the contribution of the host species on ruminal methanogens community structure.

In the present study, the ruminal archaea community in both cattle and buffaloes was dominated by the methanogens affiliated with the phylum *Euryarchaeota*. However, methanogens associated with the phylum *Crenarchaeota* were also identified that constituted <1% of total archaea in the rumen. These results are in good agreement with the previous reports [56–58], where *Crenarchaeota* methanogens were previously identified in the rumen. *Sulfolobales* are morphologically and physiologically distinguished archaea, belong to the phylum *Crenarchaeota*, and occurs mainly in extreme thermo-acidophilic ecosystems [59]. *Sulfolobales* have been previously identified in the solfataric fields all around the world. In the present study, *Sulfolobus thuringiensis* was identified in both the host species. The abundance frequency of *Sulfolobus thuringiensis* is corroborated by Dias et al. [60], who also reported that *Sulfolobus thuringiensis* are distributed up to 0.1% of total archaea in dairy calves. To the best of our knowledge, this is only the second report in the world and first from India confirming the presence of *Sulfolobus thuringiensis* in livestock. However, the methanogenic capabilities of *Sulfolobus thuringiensis*, contribution to the ruminal methanogenesis and syntrophic association with other ruminal microbes are not known and need to be explored.

Despite the greatest (72–73%) distribution of *Methanobacteriales* in the rumen, their distribution was similar in both the cattle and buffaloes. Among the genera, *Methanobrevibacter* irrespective of the host species represented the largest fraction (66–68%) of the archaea and they were similarly distributed between cattle and buffaloes. These results are in congruence

with previous studies that reported the dominance of *Methanobrevibacter* in both cattle [61, 62] and buffaloes [63, 64]. A meta-analysis [65], concluded that the methanogens associated with the genus *Methanobrevibacter* are similarly distributed in cattle and buffaloes. Further, similar distributions of *Methanobrevibacter* in Indian cattle and buffaloes belonging to two distinct geographical regions were reported in a recent study by Trivedi et al. [55]. The above studies reporting the similar proportion of *Methanobrevibacter* were either conducted separately in cattle and buffaloes or both the species were fed on different diets or having uncontrolled environmental conditions prevailing in different geographical regions; hence, these previous studies did not reveal the actual impact of the hosts on the proportion of ruminal methanogens. Our results are consistent with a previous report [52] stated the dominance of *Methanobrevibacter* in both the Jersey cattle and water buffaloes fed on a similar diet comprising corn silage and concentrate-based diet in the same environmental conditions.

At the species level, *Methanobrevibacter gottschalkii* was the most abundant methanogen and the host species did not have any significant impact on their proportion. The results on the dominance of *Methanobrevibacter gottschalkii* in the ruminal archaea community are in good agreement with previous reports [16, 66]. On the contrary, a significant difference in the abundance of *Methanobrevibacter gottschalkii* between cattle and buffaloes has been previously reported [16, 66]. This disagreement for the proportion of *Methanobrevibacter gottschalkii* between two hosts can be attributed to the confounding effect of multiple factors that remain uncontrolled in their study. Despite the limited proportion, the abundance of *Methanobrevibacter wolinii*, a hydrogenotrophic methanogen was significantly greater (P = 0.049) in cattle. Earlier studies also confirmed the presence of *Methanobrevibacter wolinii* in the rumen [67–69]. However, the contribution of *Methanobrevibacter wolinii* to ruminal methanogenesis has not yet been explored and needs further investigation to confirm the impact of proportion on the methane production between cattle and buffaloes.

*Methanomicrobiales* proportion in our study was <4% of the total archaea in both cattle and buffaloes and the abundance frequency was consistent with the global data sets [16, 56, 68]. However, few previous studies have reported an unprecedented greater abundance of *Methanomicrobium* in cattle [17, 19] or buffaloes [70–72]. This disagreement for the proportion of *Methanomicrobiales* methanogens could be attributed to the different DNA isolation methods, primer sets, and animal diets. The comparable proportion of the *Methanomicrobium* between cattle and buffaloes revealed that their proportion was somewhat consistent and not affected by the host animal under similar diet and environmental conditions.

Three species of the methanogens associated to genera *Methanocella* have been previously reported [73] and isolated from soil of rice fields [74]; however, only one species *Methanocella arvoryzae*, was identified in our study in the buffaloes rumen and that was at a very low frequency (0.002%). *Methanocellales* are non-motile, irregular rod-shaped and obligate hydrogenotrophs that primarily utilize $H_2$ as an electron donor; however, some species use formate as an electron donor. The low proportion of *Methanocellales* in the buffaloes indicated that they are unlikely to contribute significantly to ruminal methanogenesis.

*Methanomassiliicoccales* are methylotrophic methanogens that utilize methanol and methylamines for producing methane [75–77]. *Methanomassiliicoccales* were previously identified in the rumen [13, 78, 79]. Earlier, the *Methanomassiliicoccales* were placed under the RCC (rumen cluster C) and *Methanoplasmatales* [80–82]. The proportion of *Methanomassiliicoccales* in this study is consistent with Seedorf et al. [83]. At the genus level, the abundances of *Group 9*, *Group 10* and *Group 4* were similar and not affected by the host species. Jin et al. [77] from a study in goats concluded that the high grain diet feeding stimulates the greater abundances of *Group 10* and *Group 4* methanogens; while high hay diet led to a greater abundance of *Group 9* methanogens. A decrease in rumen pH due to high grain feeding is possibly a

general cause for the stimulation or suppression of a particular type of methylotrophs in the rumen [84]. Therefore, the relative abundance of *Methanomassiliicoccales* appears to be affected by the diet rather than the host. The *Methanomassiliicoccales* may have specific properties that allowed their survival or inhibition during variable pH, and this requires investigation. In our study, *Group 8* methylotrophs were exclusively identified in the cattle. The greater proportion of *Group 8* methanogens in cattle compared to their distribution in buffaloes has recently been confirmed [55]. This difference should be taken into account while developing strategies for methane mitigation from cattle and buffaloes. The methane producing capabilities of *Methanomassiliicoccales* have not yet been explored and warrants further investigation to establish their contribution to the ruminal methanogenesis.

*Methanosarcinales* grow on a broad range of substrates such as $H_2$, $CO_2$, methanol, methylamines and acetate. They are the only methanogens that possess cytochromes [85]. Methanogens associated to the *Methanosarcinales* are mostly coccoid shaped, but without motility [86] and this is the only methanogen order capable of performing acetoclastic methanogenesis [87]. In this study, although *Methanosarcinales* were distributed at very small frequencies (<0.5%) in both the cattle and buffaloes, their abundance was significantly greater (p<0.001) in cattle compared to in buffaloes. *Methanosarcinales* have been previously identified in both cattle [88, 89] and buffaloes [71]. In the present study, the low proportion of *Methanosarcinales* in both host species may be attributed to their developmental age (adult), as this group of methanogens have been reported to perform methanogenesis exclusively in the young rumen, while hydrogenotrophic methanogens are prominently involved in methanogenesis in the mature rumen [90].

In this study, the rod shape, non-motile *Methanosaeta harundinacea*, an acetoclastic methanogen was exclusively detected (0.01%) in the rumen of cattle. This methanogen was isolated from a UASB reactor [91]. Although we have not differentiated the proportion of *Methanosaeta* between solid and liquid phases in this study; nevertheless, their strict acetoclastic nature makes it likely that they are more abundant in the liquid phase [92].

On the same diet, the methane yield remains unaffected between cattle and buffaloes indicating that the methane yield is dependent on the feed rather than the species of the host. Similarly, the dominant archaeal proportion (*Methanobrevibacter*) at the genus level is also comparable between the species however, the host species level differences were observed only in the lowly abundant genus (*Methanosaeta*, *Methanocella* and *Group 8*).

## Conclusions

It is concluded that the ruminal methane yield in cattle and buffaloes fed on the same diet did not differ. Methanogens associated to the phylum *Euryarchaeota* dominated the community in both host species and *Crenarchaeota* (*Sulfolobus thuringiensis*) represented a limited fraction of the archaeal community. *Methanobrevibacter* was the most prominent genus of methanogens; however, they are distributed similarly in both host species. *Methanobrevibacter gottschalkii* despite the highest abundance did not show any host-specific difference. The relative abundances of *Methanobrevibacter wolinii* and *Methanosarcinales* were significantly greater in cattle than in buffaloes. There were a few methanogens identified exclusively either in cattle (*Methanosaeta* and *Group 8*) or in buffaloes (*Methanocella*). Thus, it is evident from the study that when the diet and environmental conditions are same, the host has a limited influence on the ruminal archaea community structure. Accordingly, we speculate that methane mitigation strategies developed in either of the hosts should be effective in the other one. However, further studies may be useful to confirm these findings with other diets and in other geographical locations.

## Supporting information

**S1 Table. Chemical composition of feed.**
(DOCX)

**S1 Fig. Rarefaction plot showing the numbers of archaeal OTUs.**
(TIF)

**S1 File. Details of reads, relative abundance and statistical analysis.**
(XLSX)

## Acknowledgments

We thank International Livestock Research Institute (ILRI), Nairobi and Indian Council of Agricultural Research, New Delhi for providing the logistics support to carry out this research under collaborative project on '*Methane Emission and its Mitigation*'.

## Author Contributions

**Conceptualization:** P. K. Malik, A. P. Kolte, R. Bhatta, H. Rahman.

**Data curation:** P. K. Malik, A. P. Kolte.

**Formal analysis:** S. Trivedi, A. Mohapatra, A. P. Kolte, V. Sejian.

**Funding acquisition:** R. Bhatta, H. Rahman.

**Investigation:** P. K. Malik, R. Bhatta.

**Methodology:** S. Trivedi, A. Mohapatra.

**Project administration:** P. K. Malik, A. P. Kolte, R. Bhatta, H. Rahman.

**Resources:** V. Sejian.

**Software:** S. Trivedi, A. P. Kolte.

**Supervision:** P. K. Malik.

**Visualization:** S. Trivedi, A. P. Kolte.

**Writing – original draft:** P. K. Malik, A. P. Kolte, V. Sejian.

**Writing – review & editing:** R. Bhatta.

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
