## [Decision Letter · Decision Letter 0]

26 May 2021

PONE-D-21-13332

Comparison of Enteric Methane Emission and Rumen Methanogens Diversity 

in Cattle and Buffaloes fed on the Same Diet

PLOS ONE

Dear Dr. Malik,

Thank you for submitting your manuscript to PLOS ONE. After careful consideration, we feel that it has merit but does not fully meet PLOS ONE’s publication criteria as it currently stands. Therefore, we invite you to submit a revised version of the manuscript that addresses the points raised during the review process.

We look forward to receiving your revised manuscript.

Kind regards,

Alex V Chaves, PhD

Academic Editor

PLOS ONE

2. Thank you for including your ethics statement:  "All the procedures involving animals were performed as per the guidelines and regulations of the Institute Animal Ethics Committee (NIANP/IAEC/1/2020/5).".   

Please amend your current ethics statement to confirm that your named ethics committee specifically approved this study.

For additional information about PLOS ONE submissions requirements for ethics oversight of animal work, please refer to http://journals.plos.org/plosone/s/submission-guidelines#loc-animal-research  

Reviewers' comments:

Reviewer's Responses to Questions

**Comments to the Author**

1. Is the manuscript technically sound, and do the data support the conclusions?

Reviewer #1: Yes

Reviewer #2: No

2. Has the statistical analysis been performed appropriately and rigorously? 

Reviewer #1: Yes

Reviewer #2: No

3. Have the authors made all data underlying the findings in their manuscript fully available?

Reviewer #1: Yes

Reviewer #2: Yes

4. Is the manuscript presented in an intelligible fashion and written in standard English?

Reviewer #1: Yes

Reviewer #2: No

5. Review Comments to the Author

Reviewer #1: This manuscript presents comparative data related to methanogenesis in cattle and buffaloes. The manuscript makes a useful contribution to this field of study. However there are several issues that need to be addressed:

1. The English expression should be substantially improved. In this regard, suggested edits are shown by track changes in the attached file.

2. Some more detail and perhaps recalculation of methane production is needed as it seems an inappropriate calculation was used. Indeed, the reported methane yields for cattle were much smaller than the usual value of about 20 g CH4/kg DMI. Recalculation of methane yields using the equation of Williams et al is likely to produce the expected values.

3. Standard errors of means or standard errors of difference should be included in Table 1.

4. A small number of statements and conclusions are not justified based on the findings of this experiment. Alternative acceptable statements are listed in the attached file.

Reviewer #2: The authors have completed an investigation into the methane emissions from both cattle and buffalo and the population demographics of methanogens in the rumen of those animals. On the surface, the experiment design looks good, but there is insufficient detail in the methods for me to assess if the experiment was done well. The low methane yields suggest there could be serious problems with the method of methane estimation.

The manuscript appears confused. I was given the impression by the introduction that if cattle and buffalo had similar methane yield and similar methanogen demographics then similar methods of mitigation could be used in both species. However, I did not see this point in the discussion which appeared to be more concerned with the function of each of the methanogens found. My understanding of the results is that both cattle and buffalo have similar methane yields and similar methanogen demographics, so methods of mitigating methane in cattle should work in buffalo. But this exciting result is not mentioned.

‘rumen fluid’ should be ‘ruminal fluid’

I suggest you use methane metrics that appear in previously published reports

methane emission – the mass of methane emitted per day, g/d

methane yield – the mass of methane emitted per unit of dry matter intake, g/kg DMI

**Title

You report that methane emission was different between cattle and buffalo due to different feed intakes. This implies that methane emission is not a suitable metric for comparison. You use methane yield in the results, so that should be used here.

**Introduction

The introduction is littered with poor English expressions. I have noted some, but others still exist.

The importance of cattle and buffalo is not explained. I can see that half the world’s buffalo are in India, but so what? You need to explain why you are focusing on cattle and buffalo.

Methane is a key part of the title and abstract but is not listed in the objectives. Given later findings, the idea of methane yield should be introduced here, then a suitable hypothesis can be made. Charmley et al 2016 (doi: 10.1071/AN15365) could be used to argue that methane yield is independent of species and feed intake.

I would like to see a hypothesis. Perhaps ‘We hypothesised that there would be no difference in the methane yield or microbial population demographics between cattle and buffalo.’

L69 The reference used is from 2016, making it 5 years out of date. Present data should be available from one of the atmospheric monitoring organisations. For example, National Oceanic and Atmospheric Administration, or CSIRO.

L69 Tg is a SI unit so does not need to be explained

L69 to 71 These statements should be referenced

L72 which country? 13% of what population of cattle?

L75 suggest ‘Methanogenesis is a major sink for hydrogen in the rumen. However, methane has an embodied energy of XXX kJ/g resulting in the methane emitted by cattle accounting for 2 – 12% of gross energy intake.’

L79 to 85 This reads as a list of things. There is no logical progression to the argument.

L87 environmental conditions cannot report anything. I suspect you mean ‘Environmental conditions have also been reported to influence the ruminal archaeal community,’

L88 Suggest ‘In a global study [20], Methanobrevibacter gottschalkii …’

Materials and methods

L107 which institute?

L116 How were the animals arranged in the shed? Williams et al 2011 (doi: 10.1071/AN15365) have shown that background concentration of gases varies within a building, even when completely open on one side.

L121 and elsewhere – I suggest using g/kg instead of parts. EG: maize grain (320 g/kg)

L123 ‘basal’ is unnecessary, there is only one diet.

L127 ‘S1 Table’ should be ‘Table S1’

L130 to 139 This description is inadequate. I cannot replicate your work from the description given. References 27 and 28 do not tell me how the technique was implemented. [27] used stainless steel spheres to sample gas for up to 6 hours, [28] does not describe an implementation of the SF6 technique. I suggest you see chapter Berndt, A, Boland, TM, Deighton, MH, Gere, JI, Grainger, C, Hegarty, RS, Iwaasa, AD, Koolaard, JP, Lassey, KR, Luo, D, Martin, RJ, Martin, C, Moate, PJ, Molano, G, Pinares-Patiño, C, Ribaux, BE, Swainson, NM, Waghorn, GC, Williams, SRO (2020) 'Guidelines for use of sulphur hexafloride (SF6) tracer technique to measure enteric methane emissions from ruminants.' (New Zealand Agricultural Greenhouse Gas Research Centre: New Zealand).

Given the animals were enclosed in a shed, details of how background gas samples were collected is critical. See Williams et al 2011 (doi: 10.1071/AN15365)

There is no indication of the extent of dilution. Were the collection vessels pressurized with nitrogen, and to what pressure. This dilution also needs to be corrected for. Not all collection vessels will be at the same residual vacuum, so the dilution will be different for each sample. For an example of how to do the correction when the canisters are still under vacuum after dilution see Moate et al 2020 (doi: 10.3390/ani10060976).

Also, the equation presented is insufficient. See equation 2 in Williams et al 2011 (doi: 10.1016/j.anifeedsci.2011.08.013) for a complete expression of the calculation.

L147 – 148. The description of the collection of ruminal fluid is insufficient to allow me to repeat the process.

L150 repeats the previous sentence.

L151 to 153 How were the samples prepared? Acidified, stained, frozen?

L158 the description of the processes are inadequate. What detector temperatures were used in your work. what carrier gas, at what flow and pressure? And what was the specification of the column? I would expect to see the model and manufacturer details.

Was the method for NH3-N exactly as per [30] or were there some implementation details that were different during this analysis?

L199 ‘vegan’ this should be identified as a version of R software.

L204 suggest ‘adonis function’ be identified as part of R

There is no mention of how the methane data were analysed.

**Results

higher is used when greater is meant.

p values – please state the actual p values in the text instead of simply thresholds.

L222 Once methane yield is defined in the introduction it can be used here. ‘However, there was no difference in methane yield between cattle (13.4 g/kg DMI) and buffaloes (13.5 g/kg DMI).’ Also, these values are much lower than those commonly reported by others.

L231 TVFA – true, but I note that the TVFA concentration in cattle is 1.6 time greater than in buffalo

L234 greater in buffalo compared to what?

L235 While I note that there is no statistical difference, the concentration of Holotrichs in cattle were only 75% of that in buffalo.

Table 1 There is no indication of error in the results but there should be

L242 ‘reads were’

Figure 2 is useless to me. So it shows OTU, but of what?

L287 p < 0.001 is sufficient.

L322 between what and buffaloes?

Figure 6 I am not able to read the text within the figure.

**Discussion

The discussion appears to be a simple restatement of results and comparisons to previous work. I did not find any new interpretation of the information nor new insights.

There needs to be an acknowledgement and discussion of the methane yield values being lower than those reported previously. There is insufficient detail in the methods for me to suggest if the values reported here are true, or the result of errors in the technique. Sampling bias, dilution correction, gas elution, GC calibration, and emission calculation are all candidates for sources of error.

L343 to 345 This is in agreement with Charmley et al 2016 (doi: 10.1071/AN15365)

L346 ‘variation’ is used when ‘difference’ is meant.

L356 ‘incredible’ is an emotional phrase. Please replace.

L354 to 363 This paragraph starts with microbiota and ends with methods. I don’t understand what point is being made here.

**References

27 has the author list duplicated

6. PLOS authors have the option to publish the peer review history of their article (what does this mean?). If published, this will include your full peer review and any attached files.

Reviewer #1: No

Reviewer #2: No

---

## [Author Response · Author response to Decision Letter 0]

16 Jun 2021

We would like to thank both the reviewers for sparing their valuable time during the COVID pandemic. The comments and suggestions provided by both the reviewers are very constructive and relevant which improved the manuscript quality substantially. This is to certify that all the suggestions have been incorporated in the revised manuscript. We are also thankful to the Editor for providing the inputs to improve the manuscript quality. The response to the reviewers and editor's comments are provided in a separate file 'Response to Reviewers' attached along with the manuscript

---

## [Decision Letter · Decision Letter 1]

9 Jul 2021

PONE-D-21-13332R1

Comparison of Enteric Methane Yield and Diversity of Ruminal Methanogens in Cattle and Buffaloes fed on the Same Diet

PLOS ONE

Dear Dr. Malik,

Thank you for submitting your manuscript to PLOS ONE. After careful consideration, we feel that it has merit but does not fully meet PLOS ONE’s publication criteria as it currently stands. Therefore, we invite you to submit a revised version of the manuscript that addresses the points raised during the review process.

Reviewer 2 has raised issues with the methodology which must be addressed before final decision. Please see the comments below. Make sure these are properly fixed.

We look forward to receiving your revised manuscript.

Kind regards,

Alex V Chaves, PhD

Academic Editor

PLOS ONE

Reviewers' comments:

Reviewer's Responses to Questions

**Comments to the Author**

1. If the authors have adequately addressed your comments raised in a previous round of review and you feel that this manuscript is now acceptable for publication, you may indicate that here to bypass the “Comments to the Author” section, enter your conflict of interest statement in the “Confidential to Editor” section, and submit your "Accept" recommendation.

Reviewer #1: All comments have been addressed

Reviewer #2: (No Response)

2. Is the manuscript technically sound, and do the data support the conclusions?

Reviewer #1: Yes

Reviewer #2: Partly

3. Has the statistical analysis been performed appropriately and rigorously? 

Reviewer #1: Yes

Reviewer #2: Yes

4. Have the authors made all data underlying the findings in their manuscript fully available?

Reviewer #1: Yes

Reviewer #2: No

5. Is the manuscript presented in an intelligible fashion and written in standard English?

Reviewer #1: Yes

Reviewer #2: Yes

6. Review Comments to the Author

Reviewer #1: The authors have appropriately addressed all my comments. There is a small number of typographical errors in the manuscript, and the authors should read the manuscript carefully and remove any remaining errors.

Reviewer #2: The methods are greatly improved and now include most of the details necessary to enable the experiment to be repeated. However, there is an issue with the calculation of methane emission and methane yield.

METHODS

Detail of the permeation tubes used is missing. How were they calibrated, and what was the mean and range of release rates? See chapter 12 of Berndt et al [11] for examples of the detail to include.

There is no indication of how the dilution by nitrogen was accounted for. Eqn 8.5 in Berndt et al. [23] should be used when there is still vacuum in the sample canisters after the addition of nitrogen. However, the dilution factors specified suggest that the sample canisters were at a positive gauge pressure after nitrogen was added. In this case Equation 8.5 should be modified such that the top line reads 101 + τf, where τf is the gauge pressure in the canister after the addition of nitrogen. Furthermore, given the research was done at 900 m elevation the 101 in the correction equation should be replaced with the local mean atmospheric pressure. This should give something like 91 + τf for the top line of the correction equation. This accounting for the dilution must be done before the calculation of methane emission. There is insufficient detail about the canister vacuum/pressure at the different stages of the process for me to determine if the dilution correction is an explanation for the low calculated methane yields (mean 13.4 g/kg DMI).

Methane yield should be calculated as (the mean methane emission over the measurement period) divided by (the mean DMI over the measurement period). The model from Williams et al. [26] can only estimate the methane yield. You have measurements so use them instead. I note that the methane yield calculated using the model from Williams et al. [26] is comparable to previous reports, so your methane yield calculated from your measurements should also be comparable.

L186 replace ‘will’ with ‘were’

L200 is 13,400 rpm or g? g should be specified.

RESULTS

L256 these are methane yield, not methane emission as specified in L255

l274 OTUs – check journal requirements. Many journals use the singular abbreviation to also mean the plural. This would mean OTU is used here.

Figure 4 The text in figure 4 is unreadable in the copy provided.

DISCUSSION

L424 which country?

GENERAL

‘Distribution’ is used when discussing the proportion of the archaea population that is occupied by a particular genera or species. I think ‘proportion’ should be used in these cases.

7. PLOS authors have the option to publish the peer review history of their article (what does this mean?). If published, this will include your full peer review and any attached files.

Reviewer #1: No

Reviewer #2: No

---

## [Author Response · Author response to Decision Letter 1]

14 Jul 2021

All the suggestions have been incorporated in the revised manuscript. The details of the changes are provided in the rebuttal letter

---

## [Decision Letter · Decision Letter 2]

22 Jul 2021

PONE-D-21-13332R2

Comparison of Enteric Methane Yield and Diversity of Ruminal Methanogens in Cattle and Buffaloes fed on the Same Diet

PLOS ONE

Dear Dr. Malik,

Thank you for submitting your manuscript to PLOS ONE. After careful consideration, we feel that it has merit but does not fully meet PLOS ONE’s publication criteria as it currently stands. Therefore, we invite you to submit a revised version of the manuscript that addresses the points raised during the review process.

We look forward to receiving your revised manuscript.

Kind regards,

Alex V Chaves, PhD

Academic Editor

PLOS ONE

Journal Requirements:

Reviewers' comments:

Reviewer's Responses to Questions

**Comments to the Author**

1. If the authors have adequately addressed your comments raised in a previous round of review and you feel that this manuscript is now acceptable for publication, you may indicate that here to bypass the “Comments to the Author” section, enter your conflict of interest statement in the “Confidential to Editor” section, and submit your "Accept" recommendation.

Reviewer #2: (No Response)

2. Is the manuscript technically sound, and do the data support the conclusions?

Reviewer #2: Yes

3. Has the statistical analysis been performed appropriately and rigorously? 

Reviewer #2: Yes

4. Have the authors made all data underlying the findings in their manuscript fully available?

Reviewer #2: Yes

5. Is the manuscript presented in an intelligible fashion and written in standard English?

Reviewer #2: Yes

6. Review Comments to the Author

Reviewer #2: The manuscript now contains all the answers to my questions. Sufficient detail is included for me to fully assess the results, and the unusual results are now explained.

I suggest a minor adjustment at line 156. Replace 'The concentrations of methane...' with 'The physical dilution with nitrogen was mathematically accounted for using the equation of Lassey et al. with adjustment for local elevation and atmospheric pressure.'

Please note that the reference at line 158 is to the wrong chapter of the guidelines. The correct reference should be

Lassey KR, Martin RJ, Williams SRO, Berndt A, Iwassa AD, Hegarty RS, et al. Analyses of breath samples. In: Lambert MG, editor. Guidelines for use of sulphur hexafluoride (SF6) tracer technique to measure enteric methane emissions from ruminants. New Zealand: New Zealand Agricultural Greenhouse Gas Research Centre; 2014. p. 166.

7. PLOS authors have the option to publish the peer review history of their article (what does this mean?). If published, this will include your full peer review and any attached files.

Reviewer #2: No

---

## [Author Response · Author response to Decision Letter 2]

24 Jul 2021

Response to the Reviewer's comments are provided in the rebuttal letter

---

## [Editor Report · Decision Letter 3]

29 Jul 2021

Comparison of Enteric Methane Yield and Diversity of Ruminal Methanogens in Cattle and Buffaloes fed on the Same Diet

PONE-D-21-13332R3

Dear Dr. Malik,

We’re pleased to inform you that your manuscript has been judged scientifically suitable for publication and will be formally accepted for publication once it meets all outstanding technical requirements.

Kind regards,

Alex V Chaves, PhD

Academic Editor

PLOS ONE

---

## [Editor Report · Acceptance letter]

2 Aug 2021

PONE-D-21-13332R3 

Comparison of Enteric Methane Yield and Diversity of Ruminal Methanogens in Cattle and Buffaloes fed on the Same Diet 

Dear Dr. Malik:

I'm pleased to inform you that your manuscript has been deemed suitable for publication in PLOS ONE. Congratulations! Your manuscript is now with our production department. 

Kind regards, 

on behalf of

Prof Alex V Chaves 

Academic Editor

PLOS ONE